# Novelty and Lifted Helpful Actions in Generalized Planning

**Chao Lei, Nir Lipovetzky, Krista A. Ehinger**

School of Computing and Information Systems, The University of Melbourne, Australia
clei1@student.unimelb.edu.au, {nir.lipovetzky, kris.ehinger}@unimelb.edu.au

## Abstract

It has been shown recently that successful techniques in classical planning, such as goal-oriented heuristics and landmarks, can improve the ability to compute planning programs for generalized planning (GP) problems. In this work, we introduce the notion of *action novelty rank*, which computes novelty with respect to a planning program, and propose novelty-based generalized planning solvers, which prune a newly generated planning program if its most frequent action repetition is greater than a given bound $v$, implemented by novelty-based best-first search BFS($v$) and its progressive variant PGP($v$). Besides, we introduce *lifted helpful actions* in GP derived from action schemes, and propose new evaluation functions and structural program restrictions to scale up the search. Our experiments show that the new algorithms BFS($v$) and PGP($v$) outperform the state-of-the-art in GP over the standard generalized planning benchmarks. Practical findings on the above-mentioned methods in generalized planning are briefly discussed.

## Introduction

*Generalized planning* (GP) studies the representation and generation of solutions that are valid for a set of planning instances from a given domain (Srivastava, Immerman, and Zilberstein 2008; Srivastava et al. 2011; Hu and De Giacomo 2011; Belle and Levesque 2016; Jiménez, Segovia-Aguas, and Jonsson 2019). Recently, Segovia-Aguas, Jiménez, and Jonsson (2019) proposed a PSPACE-complete formalism for GP problems whose solutions are *planning programs*, where a sequence of program instructions are nested with `goto` instructions such that the program can execute looping and branching structures. Candidate instructions are programmed in sequence, one line a time, while assessing whether the planning instances are solvable given a maximum number of program lines. The main algorithm that follows the heuristic search paradigm to search in the space of programs is known as Best-First Generalized Planning (BFGP) (Segovia-Aguas, Jiménez, and Jonsson 2021), where variable pointers and higher level state features allow programs to solve instances with different variables. Goal-oriented heuristics to guide BFGP have given an impressive performance to existing solvers. To further scale up search efficiency, Segovia-Aguas et al. (2022) introduced a landmark counting heuristic computed from the *fact landmarks*

extracted from each instance (Porteous, Sebastia, and Hoffmann 2001; Hoffmann, Porteous, and Sebastia 2004) and enhanced with *pointer landmarks*. A progressive search was introduced to avoid over-evaluating whether each subprogram in the search is a solution for the entire set of instances. This strategy evaluates instances incrementally, starting with a single instance, and progressively increasing the number of *active planning instances*. The combination of *Progressive heuristic search algorithm for Generalized Planning* (PGP) guided by the landmark heuristic, PGP($f_{lm}$), is the current state-of-the-art.

Besides fact landmarks, other ideas in classical planning have not been introduced to generalized planning, such as *helpful actions* (Hoffmann and Nebel 2001) and *novelty-based search* (Lipovetzky and Geffner 2012). This paper aims to define *novelty* and *lifted helpful actions* for generalized planning programs and to evaluate their performance over the standard benchmark domains. For this, we introduce novelty-based generalized planning algorithms and propose new evaluation functions beneficial for generalized planning as heuristic search. In addition, we experiment with structural program restrictions in planning programs to improve search efficiency.

## Background

The STRIPS fragment of the Planning Domain Definition Language (PDDL) (Haslum et al. 2019) describes a planning problem $P$ as $P = \langle \mathcal{D}, \mathcal{I} \rangle$ where $\mathcal{D}$ is the *domain* and $\mathcal{I}$ is an *instance*. The domain $\mathcal{D} = \langle \mathcal{F}, \mathcal{O} \rangle$ is made up of a set of *predicates* $\mathcal{F}$, and *action schemes* $\mathcal{O}$, each with a triple $\langle par, pre, eff \rangle$ where $par$ indicates *parameters* (arguments), and $pre$ and $eff$ denote *preconditions* and *effects* that are sets of predicates containing terms in $par$. The instance $\mathcal{I} = \langle \Delta, I, G \rangle$ consists of *objects* $\Delta$, initial state $I$, and goal conditions $G$, specifying the set of goal states $S_G$. $\mathcal{F}$ and $\mathcal{O}$ parameters can be instantiated with $\Delta$ resulting in a set of ground atoms $F$ and actions $O$. The classical model for planning $S^P = \langle S, s_0, S_G, A, f, c \rangle$ consists of a set of states $S = 2^F$, the initial state $s_0 = I$, the set of goal states $S_G \subseteq S$, the subset of actions $A(s) = \{a \mid pre(a) \subseteq s, a \in O\}$ applicable in $s$, a transition function $f : S \times A(s) \to S$, and the cost function $c$. A solution is a sequence of actions mapping the initial state $s_0$ into one of the goal states $s \in S_G$. We also consider other planning languages in numerical do-

mains where states are valuations over a set of numeric variables instead of predicates.

## Generalized Planning

A GP problem is commonly defined as a finite set of classical planning problems $\mathcal{P} = \{P_1, \ldots, P_T\}$, where $P_t = \langle \mathcal{D}, \mathcal{I}_t \rangle, 1 \leq t \leq T$, which belong to the same domain $\mathcal{D}$. Each instance $\mathcal{I}_t$ may differ in $I$, $G$, and $\Delta$, resulting in different $O$ and $F$. A GP solution is a program that produces a classical plan for every problem $P_t \in \mathcal{P}$.

**Planning Programs with Pointers** *Planning programs* with *pointers* $Z$, where each pointer $z \in Z$ indexes a variable/object in $\mathcal{P}$, compactly describe a scalable solution space for GP (Segovia-Aguas et al. 2022). A planning program $\Pi$, with a given maximum number of program lines $n$, is a sequence of instructions, i.e. $\Pi = \langle w_0, \ldots, w_{n-1} \rangle$, and $w_{n-1}$ is always a termination instruction, i.e. $w_{n-1} = \texttt{end}$. An instruction $w_i$, where $i$ is the location of the *program line*, $0 \leq i < n - 1$, is either: a ground planning action $a_z \in A_Z$ instantiated from $\mathcal{O}$ over $Z$, a RAM action $a_r \in A_R$ for pointer manipulation, a $\texttt{goto}$ instruction for non-sequential execution over lines, or an $\texttt{end}$ instruction. RAM actions $A_R$ include $\{\texttt{inc}(z_1), \texttt{dec}(z_1), \texttt{set}(z_1, z_2), \texttt{clear}(z_1) \mid z_1, z_2 \in Z\}$ for increasing or decreasing the value of $z_1$ by one when $z_1 < |\Delta| - 1$ or $z_1 > 0$ respectively, and setting the value of $z_2$ to $z_1$ or setting the value of $z_1$ to zero. Figure 1 illustrates the relation between ground actions $O$ and ground planning action $A_Z$, by instantiating $\mathcal{O}$ over objects $\Delta$ and pointers $Z$. The mapping between $A_Z$ and $O$ allows one $a_z$ to represent a set of ground actions $O$.

Besides $A_R$, $\texttt{test}_p(\overrightarrow{z})$ RAM actions, where $p \in \mathcal{F}$, are included over STRIPS problems to return the current *program state* interpretation of instantiated predicates $F$ over objects pointed by indices $\overrightarrow{z}$. Additionally, RAM actions $\texttt{cmp}(z_1, z_2)$ and $\texttt{cmp}_x(\overrightarrow{z_1}, \overrightarrow{z_2})$ are included in numerical domains to compare the values of two pointers $z_1 - z_2$ and the values of variables $x$ referenced by indices $\overrightarrow{z_1}$ and $\overrightarrow{z_2}$ respectively. A $\texttt{goto}$ instruction is a tuple $\texttt{go}(i', Y)$, where $i'$ is the destination line, and FLAGS $Y = \{y_z, y_c\}$ are propositions representing the *zero* and *carry* FLAGS register (Dandamudi 2005). The values of FLAGS are updated by the results of RAM actions, defined as *res*, with rules $y_z := (res == 0)$ and $y_c := (res > 0)$ to express relations, e.g. $=, \neq, <, >, \leq, \geq$. In STRIPS domains, $Y$ is set to $\{y_z\}$ alone since only Boolean logic interpretations are needed.

When $\Pi$ begins to execute on an instance $\mathcal{I}_t$, a *program state* pair $(s, i)$ is initialized to $(I_t, 0)$, where $I_t$ is the initial state of instance $\mathcal{I}_t$. Meanwhile, pointers are equal to zero, and FLAGS are set to *False*. An instruction $w_i \in \Pi$ updates $(s, i)$ to $(s', i + 1)$ when $w_i = a_z$ or $w_i = a_r$, where $s' = f(s, w_i)$ if $w_i$ is applicable, or, $s' = s$ otherwise. An instruction relocates the program state to $(s, i')$ when $w_i = \texttt{go}(i', Y)$ if $Y$ holds in $s$, or to the next line otherwise $(s, i + 1)$. $\Pi$ is a solution for $\mathcal{I}_t$ if $\Pi$ terminates in $(s, i)$ and meets the goal condition, i.e. $w_i = \texttt{end}$ and $G \subseteq s$. $\Pi$ is a solution for the GP problem $\mathcal{P}$, iff $\Pi$ is a solution for every instance $\mathcal{I}_t \in \mathcal{P}$. Figure 2 shows a fragment

of planning program $\Pi$ that can flatten a block tower with different height.

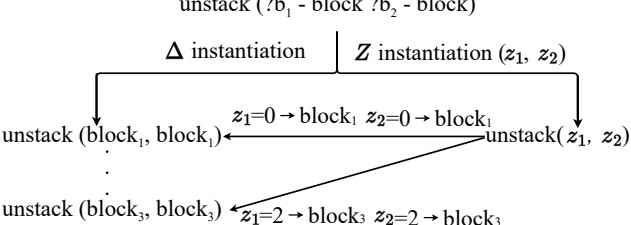

Figure 1: Example relation between action shcema $unstack$ $(?b_1 - block\ ?b_2 - block) \in \mathcal{O}$, ground actions $O = \{unstack(block_1, block_2), \ldots, unstack(block_3, block_3)\}$ over $\Delta = \{block_1, block_2, block_3\}$, and ground planning action $a_z = unstack(z_1, z_2) \in A_Z$ over $Z$. By changing the values of $z_1$ and $z_2$, $unstack(z_1, z_2)$ can remove a $z_1$ indexed block from the top of $z_2$ indexed block recursively in the *Ontable* domain.

**Generalized Planning Heuristics** Segovia-Aguas, Jiménez, and Jonsson (2021) introduced six different evaluation and heuristic functions for GP. We will employ two of them in our work. $f_1(\Pi)$ counts the number of $\texttt{goto}$ instructions in $\Pi$, and $h_5(\Pi, \mathcal{P})$ sums the Euclidean distance between the values of variables in the last reached program state and in the goals $G$ of $\mathcal{I}_t \in \mathcal{P}$. Segovia-Aguas et al. (2022) defined a landmark counting heuristic for GP over STRIPS domains extending *fact landmarks* with *pointer landmarks*. A *landmark graph* was built using the same extraction process used in LAMA (Richter and Westphal 2010), and then enriched with *pointer landmarks* indicating that each object in a *fact landmark* needs to be pointed with a pointer $z$ before the *fact landmark* is satisfied. Landmark counting heuristic, $f_{lm}(\Pi, \mathcal{P})$, guides the search by evaluating how many landmarks have to be achieved to reach the goals $G$ of $\mathcal{I}_t \in \mathcal{P}$ from the last reached program state.

**Generalized Planning Search Algorithms** Progressive GP (PGP) starts a Best First Search (BFS) with an empty program $\Pi$ of at most $n$ program lines, and the first instance as the only *active instance* (Segovia-Aguas et al. 2022). Search nodes are generated by programming up to $n$ instructions while pruning nodes recognized as dead-ends. The underlying BFS expands the best $\Pi$ in the *open* list according to its evaluation functions. PGP returns $\Pi$ as a verified solution if $\Pi$ solves all active instances and has been validated as a solution in the remaining non-active instances. If the validation fails, one of the non-active instances is added to the active instances, and the open list is reevaluated. A GP problem is unsolvable if active instances include all instances but no solution is found. PGP can trivially adapt the landmark graph by replacing $\mathcal{P}$ with active instances, and the resulting algorithm PGP($f_{lm}$) represents the state-of-the-art in GP. If all instances are active when the search starts, then PGP is equivalent to BFS.

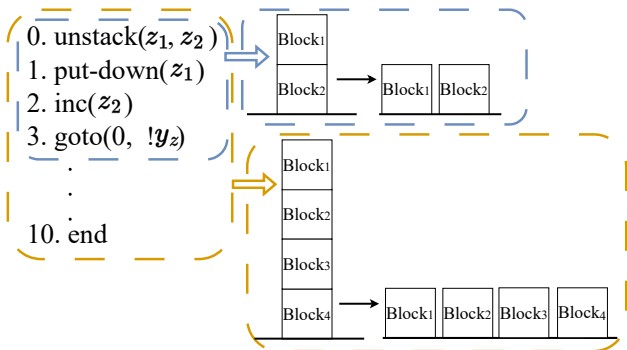

Figure 2: A fragment of planning program $\Pi$, initialized with $z_1 = 0$, $z_2 = 0$, $y_z = False$, flattens a block tower. Pointers $z_1$ and $z_2$ index object $blocks$ where $blocks = \{block_1, block_2, block_3, block_4\}$. The inner loop, line 0 to 3, tries to place the block indexed by $z_1$ on the table by increasing the value of pointer $z_2$ with the action $\texttt{inc}(z_2)$. The result of $\texttt{inc}(z_2)$ is $res = 1$ if applicable, otherwise $res = 0$ when $z_2 = 3$ resulting in $y_z = True$. The outer loop, line 4 to 10, repeatedly calls the inner loop to place top blocks on the table by iterating all block combinations, and terminates the $\Pi$ with the instruction $\texttt{end}$.

## Action Novelty in Planning Programs

The notion of *novelty* was first introduced by Lipovetzky and Geffner (2012) in classical planning to assess how novel a state $s$ is with respect to a given context $C$, defined as the states already visited by the search strategy. In classical planning, novelty is defined in terms of the predicates of a state, while, in GP, each search state is defined by the actions assigned to each line in a planning program. As a result, we define the *novelty rank* of an action $a^*$ where $a^* \in \mathcal{O} \cup A_R$, with respect to the context $C = \Pi$ of a planning program.

**Definition 1.** *The action novelty rank* $r(a^*, \Pi) = 1 + \sum_{i=0}^{n-1}[w_i = a^*]$ *is the count of the number of appearances of action* $a^*$ *in program* $\Pi$. *If action* $a^* \notin \Pi$, *then its rank is* $1$, *whereas if action* $a^*$ *appears in every line of* $\Pi$, *then its rank is* $n+1$, *where* $n$ *is the number of lines in* $\Pi$.

E.g. the action novelty rank of the action schema $visit$ given $\Pi = \langle inc(z_1), inc(z_2), visit(z_2, z_1), visit(z_1, z_2) \rangle$ is three, as $visit$, appears twice in the program. For RAM actions $inc(z_1)$ and $inc(z_2)$, the action novelty rank is two as both appear once in the program.

## Generalized Planning Novelty-based Search

In this section, we describe how to use $r(a^*, \Pi)$ in generalized planning with off-the-shelf program-based planners.

**Definition 2.** *Given a search state containing a planning program* $\Pi$, *let* $\Pi_{w_i = a^*}$ *be the planning program resulting from assigning action* $a^*$ *to the current programmable line* $i$ *in* $\Pi$. *Novelty-Based BFS, BFS(v), and Novelty-Based PGP, PGP(v), use the action novelty rank and a bound* $v$ *to prune a newly generated program* $\Pi_{w_i = a^*}$ *when* $r(a^*, \Pi) > v$.

Def. 2 considers only planning action schemas and RAM actions, e.g. $\texttt{goto}$ instruction is not a RAM action, and hence, including $\texttt{goto}$ more than $v$ times would not lead to a pruned state. This results in algorithms that encourage branching and looping in planning programs.

Action novelty rank pruning speeds up the resulting search algorithms by reducing the search space of planning programs. If $v$ equals the maximum program lines, BFS(v) and PGP(v) degrade to BFS and PGP. The crucial question is whether the resulting generalized planners can find a solution for $\mathcal{P}$ with low $v$ bounds. The answer is yes. In practice, ten out of fourteen domains can be solved with $v = 1$. The other four require $v = 2$, as in *Corridor*, *Gripper*, and *Lock* domains, the planning action *move* or RAM action $\texttt{inc}(z_1)$ is required twice in $\Pi$ to move in two directions, and in the *Fibonacci* (Fibo) domain, $\Pi$ needs two $add$ planning actions to sum the values of currently pointed variables with their previous values. The low $v$ bounds are a result of the special structure of planning programs with pointers, where a single action schema can represent multiple ground actions through instantiations over the pointers $Z$. Branching and looping allow $\Pi$ to change $Z$ to different values with the least possible number of RAM actions. Restricting the search of programs to highly ranked novel actions ensures that the resulting $\Pi$ reuses planning and RAM actions to generate different action effects to solve all the planning instances.

## Lifted Helpful Actions

*Helpful actions* (HA) were first introduced in the context of classical planning and played a key role in several state-of-the-art planners (Hoffmann and Nebel 2001; Helmert 2006; Richter and Westphal 2010; Lipovetzky and Geffner 2017). HA are the subset of applicable actions that appear in a delete-relaxed plan, computed for every expanded state in order to reduce the branching factor. Instead, we compute *once* all the *lifted helpful actions* in an instance $\mathcal{I}_t \in \mathcal{P}$ of a GP problem. Let $L(C) = \{l \mid l(par) \in C \subseteq \mathcal{F}\}$ be the function that lifts the representation of a collection $C$ by removing arguments of its predicates. We define $U_0 = L(G \setminus I)$ as the first lifted unachieved layer of ground predicates. The lifted HA are then defined as the action schemas $S_i = \{a \mid L(eff(a)) \cap U_i \neq \emptyset, L(eff(a)) \cap U_j = \emptyset, 0 \leq j < i, a \in \mathcal{O}\}$ that support the lifted unachieved predicates unsupported in previous layers. We then update the lifted unachieved predicates $U_{i+1} = \{U_i \cup (L(pre(a)) \setminus L(I)) \setminus L(eff(a)) \mid a \in S_i\}$, for $i \geq 0$, computing the regression over the lifted HA. Once two consecutive layers of lifted unachieved predicates are the same, i.e. cannot be updated further, then the computation terminates. The set of lifted HA is $H = \bigcup_i^n S_i$, where $n$ is the final layer where the computation converged. This computation is similar to backward reachability over the lifted representation. The lifted HA for $\mathcal{P}$ are set to be the union over $H$ for each instance in $\mathcal{P}$. Intuitively, the reachable lifted actions that support the unachieved goals of an instance are deemed as helpful. We will refer to lifted helpful actions as helpful actions.

## Heuristics, Costs and Structural Restrictions

We introduce three new evaluation functions to exploit the structure of planning programs: $f_{ha}(\Pi, \mathcal{P})$ is the number

| | $n/|Z|$ | $B_{5,1}$/(Improved) $B_{5,1}$ | | | $P_{5,1}$/(Improved) $P_{5,1}$ | | |
|---|---|---|---|---|---|---|---|
| | | T | Ex | Ev | T | Ex | Ev |
| Fibo | 7/2 | 52/**30** | 43K/**30K** | 1M/**0.7M** | 18/**13** | 43K/**30K** | 1M/**0.7M** |
| Find | 6/3 | 21/**8** | 53K/**18K** | 0.8M/**0.3M** | 17/**7** | 55K/**18K** | 0.8M/**0.3M** |
| Reverse | 7/2 | 138/**56** | 0.5M/**0.2M** | 9M/**3M** | 70/**27** | 0.5M/**0.2M** | 9M/**3M** |
| Sorting | 8/2 | 2K/**741** | 5M/**2M** | 108M/**40M** | 1K/**441** | 5M/**2M** | 107M/**40M** |
| Select | 6/2 | 4/**1** | 19K/**5K** | 0.3M/**76K** | 2/**1** | 19K/**5K** | 0.3M/**76K** |

Table 1: Comparisons of improved BFS and PGP with their original versions. B and P are acronyms of BFS and PGP respectively; $n$ stands for the number of program lines; $|Z|$ stands for the number of pointers; T is the total time in seconds; Ex is the number of expanded nodes, and Ev is the number of evaluated nodes (K is $10^3$ and M is $10^6$). Best results are in bold.

of planning actions in $\Pi$ that are not helpful actions for $\mathcal{P}$; $f_{ln}(\Pi)$ is the number of instructions in $\Pi$ except `goto`, `test`, and `cmp`; $f_{cn}(\Pi, \mathcal{P})$ is the number of yet to be tested ground atoms or compared ground atom pairs by the action `test` or `cmp` respectively, calculated by executing $\Pi$ on each instance in $\mathcal{P}$. All these functions are *cost functions*, so smaller values are preferred.

$f_{ln}(\Pi)$ is designed to allow as many branching and looping operations, not increasing the cost of $\Pi$ when it contains `goto`, `test` and `cmp`. A similar idea prioritizing programs with the maximum number of loops has been explored by Segovia-Aguas, E-Martín, and Jiménez (2022). $f_{cn}(\Pi, \mathcal{P})$ encourages $\Pi$ to explore new states during the search in order to test or compare as many ground atoms or atom pairs as possible. These three evaluation functions can be used together in BFS($v$) and PGP($v$), by replacing $\mathcal{P}$ with active instances, since $f_{ha}(\Pi, \mathcal{P})$ and $f_{ln}(\Pi)$ can be computed in linear time, and $f_{cn}(\Pi, \mathcal{P})$ is linear on the longest execution of $\Pi$ over $\mathcal{P}$.

We adopt two structural restrictions over the space of programs to keep the search space tractable without sacrificing completeness: 1) RAM actions `clear`, `dec` and `set` are not allowed in the first line, as they induce an unnecessary initial search plateau over the pointers, and 2) the destination line of a `goto` instruction is not allowed to be another `goto`. One `goto` instruction can represent the same logic.

## Evaluation

Segovia-Aguas, Jiménez, and Jonsson (2021) and Segovia-Aguas et al. (2022) introduced eight STRIPS domains and six numerical domains as generalized planning benchmarks. We strictly followed their training and validation requirements in our experiments. The numerical domain *Triangular Sum* (T.Sum) includes a `test` action to express the goal condition. The combination of $(f_{lm}, f_1)$, and $(h_5, f_1)$ are used in BFS and PGP to serve as baselines, where the search is guided by the first term and breaks ties with the second (Segovia-Aguas et al. 2022). For evaluation functions with three heuristics, the evaluation function breaks ties lexicographically. Landmarks are only applied over STRIPS domains since, except *Fibo* and *T.Sum*, actions in other numerical domain benchmarks are precondition-free. All experiments were conducted on a cloud computer with clock speeds of 2.45 GHz EPYC processors and processes time or memory out after 1 hour or 8 GB.

## Synthesis of GP Solutions

Table 1 reports the performance of improved BFS and PGP that apply the structural restrictions compared with their original versions where $h_5$ and $f_1$ are represented by their subscripts. We only illustrate domains with significant improvements. All omitted domains have an improvement of $1\%$ to $15\%$. The domains in Table 1 benefit greatly from structural program restrictions. Planning actions $A_Z$ programmed early in $\Pi$ to achieve sub-goals improve $h_5$ ability to guide the search. As a result, we keep these restrictions in the remaining experiments.

Table 2 summarizes the performance of BFS($v$) and PGP($v$) in STRIPS domains, the upper part of the table, and numerical domains, the lower part of the table, over six evaluation and heuristic function combinations represented by their subscripts. We only use $f_{ha}$ in STRIPS domains, same as for $f_{lm}$, as numeric domains are precondition free, while $f_{cn}$ and $f_{ln}$ are used in numerical domains, where `cmp` and `goto` are essential to express condition check, looping and branching. To save space, we only display the combinations that give the best performance. In Table 2, BFS($v$) and PGP($v$) match or outperform BFS and PGP in terms of search time expanded, and evaluated nodes in every domain, showing the effectiveness of novelty rank pruning. The new evaluation functions show strong efficiency in some domains when integrated with previous functions $f_1$, $f_{lm}$, and $h_5$.

BFS($v$)$_{5,1}$ outperforms BFS$_{5,1}$ in all domains. When $f_{lm}$ and $f_1$ are used, this advantage remains except in the *Intrusion* domain where BFS($v$) and BFS have the same performance. BFS($v$)$_{lm,1,ha}$ is slightly time inefficient compared with BFS($v$)$_{lm,1}$ in the *Gripper* domain but still better than the baseline BFS$_{lm,1}$. BFS($v$)$_{5,ln}$ displays a considerable jump of performance in the *Sorting* domain, while it is deficient in the *Fibo* domain. BFS($v$)$_{5,1,cn}$ is less efficient in the domain *Sorting* compared with BFS($v$)$_{5,ln}$, while it maintains efficiency gains in all numerical domains compared with the baseline BFS$_{5,1}$. BFS($v$)$_{5,cn,1}$ dominates in *Select* and *Sorting* domains, reducing the search time from 741s to 1s, as $f_{cn}$ is encouraging `cmp` to be programmed at the first line of $\Pi$, which reduces the search space significantly. At the same time, it is slightly inefficient in the *Fibo* domain.

PGP($v$)$_{5,1}$ solves one more domain, *Spanner*, than the baseline and performs better among all solved domains compared with PGP$_{5,1}$. It reveals the best result in *Corridor*, *Gripper*, *Fibo*, *Reverse* and *T.Sum* domains. PGP($v$)$_{lm,1}$ dominates in the domain *Lock* and displays a significant improvement in *Baking*, *Ontable*, and *Spanner* compared with PGP$_{lm,1}$. PGP($v$)$_{lm,1,ha}$ improves the results further and dominates all other methods in *Baking* and *Spanner*; besides, it expands and evaluates the least number of nodes in the *Ontable* domain. PGP($v$)$_{5,ln}$, PGP($v$)$_{5,1,cn}$ and PGP($v$)$_{5,cn,1}$ reveal the same strengths and weaknesses as their BFS($v$) versions in numerical domains. The relation between BFS and PGP and between landmarks and $h_5$ in STRIPS domains have been discussed by Segovia-Aguas et al. (2022).

| | $n/\|Z\|/v$ | $B_{5,1}$ | | $B(v)_{5,1}$ | | $B_{lm,1}$ | | $B(v)_{lm,1}$ | | $B(v)_{lm,1,ha}$ | | $P_{5,1}$ | | $P(v)_{5,1}$ | | $P_{lm,1}$ | | $P(v)_{lm,1}$ | | $P(v)_{lm,1,ha}$ | |
|---|---|---|---|---|---|---|---|---|---|---|---|---|---|---|---|---|---|---|---|---|---|
| | | T | Ex/Ev | T | Ex/Ev | T | Ex/Ev | T | Ex/Ev | T | Ex/Ev | T | Ex/Ev | T | Ex/Ev | T | Ex/Ev | T | Ex/Ev | T | Ex/Ev |
| Baking | 13/6/1 | - | - | ○ | ○ | - | - | - | - | - | - | ○ | ○ | ○ | ○ | 72 | 30K/0.9M | 3 | 501/20K | **2** | **500/20K** |
| Corridor | 10/2/2 | 41 | 3K/60K | 19 | 2K/45K | 108 | 22K/0.4M | 60 | 17K/0.3M | 59 | 17K/0.3M | 5 | 3K/59K | **4** | **2K/45K** | 37 | 19K/0.3M | 25 | 14K/0.2M | 25 | 14K/0.2M |
| Gripper | 8/4/2 | 5 | 2K/53K | 4 | **2K/52K** | 48 | 19K/0.3M | 28 | 18K/0.3M | 34 | 20K/0.4M | **1** | 2K/53K | **1** | **2K/52K** | 10 | 19K/0.3M | 10 | 18K/0.3M | 11 | 20K/0.4M |
| Intrusion | 9/1/1 | 54 | 44K/0.8M | 22 | 27K/0.4M | **0** | **8/188** | **0** | **8/188** | **0** | **8/188** | 14 | 44K/0.8M | 8 | 27K/0.4M | **0** | **8/188** | **0** | **8/188** | **0** | **8/188** |
| Lock | 12/2/2 | - | - | ○ | ○ | - | - | - | - | - | - | ○ | ○ | ○ | ○ | **3** | 1K/26K | **3** | **1K/25K** | ○ | ○ |
| Ontable | 11/3/1 | - | - | - | - | - | - | - | - | - | - | 24 | 9K/0.3M | 9 | 6K/0.2M | 308 | 3K/0.1M | 87 | 1K/52K | 42 | **685/24K** |
| Spanner | 12/5/1 | - | - | - | - | - | - | - | - | - | - | - | - | 824 | 0.3M/8M | 178 | 23K/0.6M | 77 | 7K/0.2M | **76** | **7K/0.2M** |
| Visitall | 7/2/1 | **0** | 44/489 | **0** | 18/211 | 8 | 25/239 | 2 | **18/158** | 2 | **18/158** | **0** | 81/1K | **0** | 18/211 | **0** | 51/448 | **0** | **18/158** | **0** | **18/158** |

| | $n/\|Z\|/v$ | $B_{5,1}$ | | $B(v)_{5,1}$ | | $B(v)_{5,ln}$ | | $B(v)_{5,1,cn}$ | | $B(v)_{5,cn,1}$ | | $P_{5,1}$ | | $P(v)_{5,1}$ | | $P(v)_{5,ln}$ | | $P(v)_{5,1,cn}$ | | $P(v)_{5,cn,1}$ | |
|---|---|---|---|---|---|---|---|---|---|---|---|---|---|---|---|---|---|---|---|---|---|
| | | T | Ex/Ev | T | Ex/Ev | T | Ex/Ev | T | Ex/Ev | T | Ex/Ev | T | Ex/Ev | T | Ex/Ev | T | Ex/Ev | T | Ex/Ev | T | Ex/Ev |
| Fibo | 7/2/2 | 30 | 30K/0.7M | 28 | 26K/0.6M | 96 | 0.3M/5M | 27 | 33K/0.8M | 37 | 69K/1M | 13 | 30K/0.7M | **11** | **26K/0.6M** | 65 | 0.3M/5M | 13 | 33K/0.8M | 19 | 69K/1M |
| Find | 6/3/1 | 8 | 18K/0.3M | 6 | 13K/0.2M | 3 | 3K/54K | 6 | 12K/0.2M | 3 | **2K/38K** | 7 | 18K/0.3M | 5 | 13K/0.2M | 3 | 4K/55K | 5 | 12K/0.2M | **2** | 2K/39K |
| Reverse | 7/2/1 | 56 | 0.2M/3M | 22 | 0.1M/2M | 22 | 0.1M/2M | 21 | 0.1M/2M | 21 | 0.1M/2M | 27 | 0.2M/3M | **11** | **93K/2M** | **11** | 94K/2M | **11** | 93K/2M | **11** | 93K/2M |
| Sorting | 8/2/1 | 741 | 2M/40M | 222 | 0.6M/13M | 46 | 74K/2M | 185 | 0.5M/12M | **1** | **2K/52K** | 411 | 2M/40M | 140 | 0.6M/13M | 25 | 74K/2M | 118 | 0.5M/12M | **1** | 2K/55K |
| Select | 6/2/1 | 1 | 5K/76K | **0** | 4K/52K | **0** | 1K/18K | **0** | 3K/49K | **0** | 683/1K | 1 | 5K/76K | **0** | 4K/53K | **0** | 2K/20K | **0** | 3K/50K | **0** | 746/10K |
| T.Sum | 6/2/1 | 9 | 11K/0.2M | 5 | 7K/0.1M | 5 | 7K/0.1M | **4** | 7K/0.1M | 6 | 10K/0.2M | 8 | 11K/0.2M | **4** | **7K/0.1M** | 5 | 7K/0.1M | **4** | 7K/0.1M | 5 | 10K/0.2M |

Table 2: Results over BFS($v$) and PGP($v$) with different evaluation and heuristic function combinations. $v$ is the bound of the action novelty rank; symbols **-** and ○ denote time and memory exceeded. Other metrics are the same in Table 1. Best results are in bold.

## Discussion

The action novelty rank $r(a^*, \Pi)$ improves BFS and PGP by adding a restriction on action occurrences in $\Pi$. Helpful actions guide the search with $f_{ha}$ to avoid considering programs in the search with irrelevant actions. For example, in the *Ontable* domain, the action *stack* is irrelevant since it is not a helpful action, only $putdown$ is part of a valid planning program. On the other hand, helpful actions may misguide the search when necessary actions are absent due to the open-world assumption over $G$. For example, in the *Lock* domain, the action *move* is ignored in helpful actions since the goal state only contains the unachieved predicate *unlock*, and the only helpful action is *open*. We experimented with the restriction that planning actions can be programmed only when applicable. In *Corridor*, *Ontable*, and *Spanner*, solutions cannot be found as extra lines are required to update the object pointers until `test` actions return true for all ground atoms in the precondition of planning actions. Evaluation functions $f_{ln}$ and $f_{cn}$ encourage $\Pi$ to build a complex program logic by including instructions `goto` and `cmp` that are in line with generalized planning usage scenarios. They are influential in numerical domains *Find*, *Sorting*, and *Select*.

## Conclusion

We showed that structural program restrictions improve the performance of GP, and action novelty rank scales up GP algorithms significantly over all the domains with a bound of $v = 1$ or $v = 2$. We proposed a characterization of lifted helpful actions in GP and experimented with different evaluation function combinations using new functions $f_{ha}$, $f_{ln}$, and $f_{cn}$. Other lifted HA extraction methods (Corrêa et al. 2021; Wichlacz, Höller, and Hoffmann 2022) and novelty-based search strategies (Lei and Lipovetzky 2021; Singh et al. 2021; Corrêa and Seipp 2022) proposed for classical planning could be adopted by research on GP as heuristic search.

## Acknowledgements

Chao Lei is supported by Melbourne Research Scholarship established by The University of Melbourne.

This research was supported by use of The University of Melbourne Research Cloud, a collaborative Australian research platform supported by the National Collaborative Research Infrastructure Strategy (NCRIS).

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
