# OpenReview forum: "Novelty and Lifted Helpful Actions in Generalized Planning"
_icaps-conference.org/ICAPS/2023/Workshop/HSDIP — ICAPS HSDIP 2023_

### Official Review · Reviewer_3e2t · 2023-04-24
**Review of Novelty and Lifted Helpful Actions in Generalized Planning - Accept**

**Rating:** 7
**Confidence:** 4

**Review:**

The paper is based on a new line of work in generalized planning that generates solutions via heuristic search in an assembler-like program format. The contributions of the paper are 1) a technique to prune states based on a new notion of novelty for generalized planning, 2) several new evaluation functions including one based on helpful lifted actions, and 3) an introduction of structural restrictions to reduce the search space while maintaining completeness. All techniques rely on a fairly simplistic view of the task and solution structure. Lifted actions are only considered in the pure lifted form, without analyzing any corresponding grounded actions. Program lines are assessed in a line-by-line basis, without analyzing their relationships to other program lines. In my opinion, this raises concerns about the scalability of this approach to more complex program structures. Though, as developing solutions of this form is still in an early stage, it seems reasonable to use this as a starting point.

The write-up is well presented and mostly easy to follow. The experimental evaluation shows slight improvements. I believe the paper is a fitting contribution for HSDIP and recommend its acceptance.

I have some questions/suggestions for the authors that I will list below:

* Have you compared a blind search to an improved novelty-based blind search? I'm interested in this to determine if the heuristics already offer similar guidance to the novelty-based improvement search.
* Can you share insights on the conceptual link between evaluation functions based on lifted helpful actions and landmarks? I believe that lifted helpful actions should be reflected in landmarks in some way.
* In your experimental evaluation, you focus exclusively on approaches within the same heuristic search framework. I suggest including Segovia-Aguas, Jiménez, and Jonsson's (2019) compilation-based approach as a reference. Since the representation was described to be hard to ground, it may be reasonable to combine it with a lifted search configuration based on Corrêa et al. (2020), unless there are practical constraints. Possible configurations could involve computing hadd/hff with helpful actions (Corrêa et al. 2021, 2022), landmarks (Wichlacz, Höller and Hoffmann 2022) or novelties (Corrêa and Seipp 2022).
* I would suggest contrasting your definition of novelties to their classical counterparts. I think that it is important to highlight that your definition does not capture the original definition's emphasis on unique combinations of facts (or program lines in your context).

Typo:
* "where each pointer [z ⊆ Z] indexes a variable/object" -> z \in Z

---

> ### Author Response · Authors · 2023-05-01
> **Response to Reviewer**
>
> We really appreciate all the detailed comments, and we will follow the advice and clarify all your points and questions in the final version.
>
> Q1: We aim to run an experiment and include a comment about the effect of novelty and heuristics individually in our final version. Thanks for pointing this out.
>
> Q2: There is an interplay between helpful actions and landmarks. The landmark counting heuristic encourages the search to produce planning programs that achieve landmarks. Helpful Actions can assist the search to reach landmarks with actions that are likely to be part of a valid solution, hence, improving the landmark counting value of the generated programs. For example, it can promote programs that achieve the landmark $handempty$ with action $putdown(x)$ rather than $stack(x,y)$ in the $Ontable$ domain for block $x$ whose target position is not on top of block $y$.
>
> Thanks for your suggestions. We will clarify and acknowledge Segovia-Aguas, Jiménez, and Jonsson's (2019) work accordingly in the final version. Combining with other lifted search approaches, such as the one proposed by Corrêa et al. (2020), and with hadd/hff (Corrêa et al. 2021, 2022) are promising directions for future work. We will mention this in the final version. For novelty, We will motivate further the connection with classical planning novelty, novelty features, and program novelty and emphasize the difference between our definition and the original definition in the final version. Thanks!

---

### Official Review · Reviewer_zmtH · 2023-04-26
**Interesting addition to generalized planning based on planning programs through the inclusion of action novelty, lifted helpful actions, a few heuristics and structural restrictions of the search space. Background and level of detail of the new techniques could be expanded, as well as the discussion of experimental results.**

**Rating:** 7
**Confidence:** 3

**Review:**

This paper deals with generalized planning (GP), and in particular,
with planning programs for solving GP problems. It introduces action
novelty rank, which is the number of occurrences of an action in a
program that can be used to prune a newly generated program if the last
added action has an action novelty rank larger than a given bound. The
paper also defines lifted helpful actions as those action schemas that
add a lifted predicate (lifted from ground predicates) from a given
collection of predicates for the first time (i.e., reachability
analysis style of computation). Helpful actions are used as a heuristic
preferring programs with fewer non-helpful actions. Other additions are
heuristics minimizing the number of instructions or the number of
untested ground atoms. Finally, the paper suggests two simple
structural restrictions of the space of programs. Experiments show that
the latter are always helpful in progressive search for GP (PGP) and
best-first search (BFS). Furthermore, many combinations of adding the
new heuristics to existing ones improve the performance of both search
algorithms.

The topic is clearly relevant to the workshop. The suggested additions
of generalized planning as planning programs are interesting and
increase performance.

While the paper is mostly well written, I found the formal presentation
of GP and planning programs less easy to understand due to the
presentation being very condensed, sometimes missing to define certain
notions. For instance, I assumed the set of lifted predicates F to be a
set of symbols, rather than the full-fledged predicates including
parameters, which I had to deduce from L(C) = {l | l(par) \in F
\subseteq F} much later (where l(par) is undefined notation). Also,
"par" was never explained to be parameters, neither for predicates nor
actions. States are never defined. The cost function c is not part of a
problem P. The definition of a solution says that it "maps" the initial
state into a goal state, which doesn't even mention the transition
function. At many places, the text uses letters such as O instead of
spelling out "actions", e.g., "resulting in different O and F". This is
not straightforward to read if one doesn't have the definition of D and
I_t in mind. The term variable is used once without explaining if that
refers to variables used as parameters of lifted structures or rather
refer to ground instantiated predicates. The term "program state" is
used ahead of its definition. Notation like "eff(a)" is not defined.
While all of this might seem nitpicky, I think it should be addressed
for a better reading experience. More generally, since GP is not a
topic everyone is familiar with, adding an example would be very
helpful. In particular, I assume that for the paper to be well-received
at SoCS, the concept of GP and planning programs should be explained in
more detail and include an example.

Regarding content, I like the suggested additions, but wondered if they
are connected in any way. It seems like all of the suggested techniques
are orthogonal to each other, which  makes reading the different three
main sections a bit disconnected from each other.

The experiments are discussed only very briefly and do not present any
insights as to why certain techniques or combinations of heuristic work
better than others generally and on specific domains. Also, the
discussion basically doesn't refer to specific results in the table,
which could therefore as well have been omitted at all. (Of course, I'm
not suggesting to remove the data, but to discuss it more.) It would be
great to expand the discussion and provide any insights and conclusions
to be drawn from the results other than that the presented techniques
"work".

Furthermore, I wondered if the structural restrictions could be harmful
in any situation. From the experiments, it seems like this is not the
case and they should always be included.

Overall, the paper clearly meets the requirements for HSDIP and should
be accepted.

Minor comments:
- plural of index is indices, not indexes
- w(a, \Pi) -> r(a, \Pi)
- "as many branching and looping operations": as many as what? as
possible?
- Segovia-Aguas, Jiménez et al.: wrong citation style; either
"first-author et al. (year)" or "author, author, author ...(year)"
- the notation B_x,y to hint at algorithm BFS combined with h_5 and f_1
is not ideal; the numbers could be taken for anything instead of h and
f. (In general, naming a heuristic "h_5" seems like a bad idea.)
- "in the camera-ready." -> in the camera-ready paper.

---

> ### Author Response · Authors · 2023-05-01
> **Response to Reviewer**
>
> Thank you for your detailed comments and for acknowledging the contributions of the paper.
>
> We will improve the description of GP and give an example in the final version.  We will carefully use and clarify terms in the final version to avoid any confusion. Besides, we will follow your advice and give a more insightful analysis of the experiment results. For structural restrictions, We did not observe any drawbacks in our experiments. The structural restrictions are tightly related to symmetry breaking in classical planning, and all these restrictions would not break the completeness of algorithms.

---

### Decision · Program_Chairs · 2023-05-05

**Decision:**

Accept

**Comment:**

We are happy to announce that the paper is accepted to HSDIP 2023.

Please make sure to address the reviewers' comments in the final version.